# A Psychometric Pilot Study Examining the Functions of Suicidal Communications Using IRT and Factor Analysis

**DOI:** 10.3390/ijerph191610081

**Published:** 2022-08-15

**Authors:** Kaitlyn R. Schuler, Margaret M. Baer, Ryon C. McDermott, Phillip N. Smith

**Affiliations:** 1Department of Psychology, University of South Alabama, Mobile, AL 36608, USA; 2Department of Psychology, University of Toledo, Toledo, OH 43606, USA

**Keywords:** suicidal communications, suicide disclosure, ambivalence, functions, motivations

## Abstract

Background: Suicide prevention, an important public health issue, relies on suicidal communications to identify and intervene with those at risk. Scant research tests explicit theories of suicidal communication impeding applications to prevention science. The current study pilots a new measure assessing the functions of suicidal communications using factor analysis and item response theory. Methods: MTurk workers (*n* = 898) completed an anonymous survey. The original scale included 35 items refined using exploratory and confirmatory factor analysis, bifactor modeling, and item response theory. Results: The initial EFA identified a two-correlated-factor solution. The two-correlated-factor and unidimensional models yielded a poor fit. A bifactor model yielded a borderline to acceptable fit. The final four items were identified using a bifactor model and item response theory graded response models capturing ambivalence resolution defined as behaviors aimed to aid in suicide decision making. The final model yielded an excellent fit: 𝝌^2^(2) = 1.81, CFI (1.00), TLI (1.00), RMSEA (0.00), and SRMR (0.01). Conclusions: There may be one function of suicidal communications. Disclosure may elicit connection and reasons for living that serve as barriers to suicide and resolve ambivalence. Key limitations include convenience sampling and limited validity measures. Future research should partner with participants to improve scale and theory development efforts.

## 1. Current Research

Suicide prevention is an urgent public health problem in the U.S. [1]. In comparison with years prior, past-year prevalence rates of suicide ideation (4.3%) and attempts (0.6%) remain elevated [1]). Suicide prevention and intervention efforts depend on individuals at risk communicating to others their ideation or intent. Even suicide prevention efforts that rely on passive detection (e.g., machine learning) ultimately require that an individual at risk for suicide communicates with health care professionals or social networks for any form of intervention to take place. Despite the importance of understanding, facilitating and appropriately responding to suicidal communications, this area of research remains relatively untapped [2]. 

The largest body of literature on this topic focuses on prevalence rates. A recent meta-analysis indicated that 44.5% of decedents were known to have communicated about their intent before death [3]). Across studies using large samples, there is high variability in rates. Between 33% and 61% of individuals with a past suicide attempt reported communicating about their intent [4,5,6]. Although suicidal communications are not homogenous [3,6], communication is often assumed to be direct and verbal and serve a help-seeking function. Prior research suggests [6,7] that there are various forms and functions of suicidal communications. However, current research largely neglects the potential for a range of forms and functions of suicidal communications, which may explain the high variability in prevalence across samples. 

This neglect of the complex, multifaceted nature of suicidal communications is directly related to the gap in measurement. To date, suicidal communication measurement primarily consists of single-item and unvalidated interviews [8,9]. Only two published multi-item measures exist related to suicidal communications. The validated Self-Harm and Suicide Disclosure Scale (SHSDS) [10] assesses the level of detail attempt survivors provide (e.g., method used and severity). Hypothesized to be related to suicide stigma, this unidimensional measure assesses how much information survivors share with others about their suicide ideation and attempt histories. The Family Quality Reaction Scale (FQRS) [11] assesses the helpfulness of communication recipients’ reactions. The scale is unidimensional and captures increased comfort and perceived helpfulness following disclosure. These measures are a promising initial attempt at comprehensive measurement of suicidal communications. Frey and colleagues [7,10,11] have provided the most significant contribution thus far to this area of research. Two limitations necessitate creating additional measures: these measures do not exist within a theoretical model, and do not completely address the need for a comprehensive examination of suicidal communications. While it is helpful to understand the amount of detail provided in suicide disclosure and in the helpfulness of reactions, these measures do not work together to describe the process of suicidal communications. A comprehensive and theoretically based battery of validated instruments would fill this critical gap. 

The current article presents a measure of the functions of suicidal communications grounded in a preliminary empirical framework (see Figure 1. This theory states that suicidal communications are an interpersonal construct that must be defined in terms of their form, function, and perceived reactions as suggested by prior research [6,7]. The current article aims to fill a gap in measurement of the functions construct. Prior theoretical work exists to hypothesize multiple functions of suicidal communications [7]). However, this hypothesis has not been tested empirically through factor analytic methods. First, such communications may serve a help-seeking function (e.g., treatment seeking and support seeking). For instance, in a series of qualitative interviews about suicide communication with a small sample of 40 individuals with a history of suicide attempts, the majority endorsed getting help as the primary reason to communicate [12]. Although the term ‘getting help’ was broadly used, examples consisted of seeking care from any health care provider or paraprofessional and obtaining support or help from friends or family to get connected with care. Similarly, in an undergraduate sample with past suicidal ideation who reported being directly asked by others if they were considering suicide, 40.6% reported communicating to get support in seeking treatment, 36.3% reported communicating to receive medication, 33.8% to receive therapy, and 11.9% to seek hospitalization [9]. These studies suggest that help seeking is likely a commonly endorsed function of suicidal communications. 

Individuals may also communicate about suicide to plan and prepare for suicide, such as directly soliciting another’s assistance to attempt suicide, obtaining another person’s assistance to prepare for death, or communicating as a way of obtaining closure on a relationship before death (e.g., saying goodbye to loved ones and resolving interpersonal disputes). Direct solicitation of others’ assistance to die by suicide has most frequently been examined on pro-suicide message boards and chatrooms, and other discussion-based mechanisms [13]. This form of communication may include a person requesting and receiving detailed information about suicide methods or general self-injury techniques [14,15], or receiving encouragement after communicating intent [16,17,18,19] In one study, 14.5% of internet forum users reported using forums specifically for these reasons [20]. Although no literature on tying up loose ends before death [21] directly speaks to others’ involvement, some preparations for death (e.g., arranging care for dependents) are highly likely to involve a trusted person. Finally, given high percentages of saying goodbye to loved ones found in other modes of death (physician-assisted death; 87%) [22], some individuals may similarly say goodbye before suicide. For instance, 38% of a male sample who had attempted suicide reported saying goodbye to loved ones before the attempt [23]. Suicidal communications may therefore be motivated by plans and preparations for suicide. 

Suicidal communications may also serve an intrinsic interpersonal emotion regulation function. Broadly, intrinsic interpersonal emotion regulation occurs when individuals seek and engage in specific interpersonal interactions (e.g., venting, reassurance seeking, and communicating emotions) to regulate their emotions [24,25]. Specifically, as thoughts of suicide may be highly distressing, negative affect may be alleviated by communicating with others. Research highlighting reported positive feelings from individuals after communicating about or discussing suicide during a research study e.g., [26,27,28] and crisis line call [29] supports this hypothesis. Indeed, in an undergraduate sample who had been directly asked by others if they had thoughts of suicide, 69% endorsed disclosing to obtain emotional support [9]. In the qualitative interview study previously mentioned, participants endorsed both the above—sharing their suicidal ideation/experiences to process with others and strengthening their relationship with that person [12]. Ample evidence exists to support a potential interpersonal emotion regulation function of suicidal communications. 

Finally, suicidal communications may be motivated by a desire to resolve ambivalence about suicide. Research on the decision-making process involved in suicide attempts highlights a period of “mulling over” [30] that may encompass last attempts at seeking help, conflict resolution, and steps to relieve psychological pain. Approximately half of a psychiatric sample reported a consideration period (from first current thought to attempt) of less than 10 minutes and those who took longer reported elevated suicidal intent [31]. Indeed, increased time for consideration may indicate greater chronic suicide risk. Attempts at ambivalence resolution through suicidal communications may, therefore, also represent heightened chronic risk. 

## 2. Current Study

The current measure draws on prior literature to examine the multiple functions hypothesis and fits within a preliminary empirical framework (see Figure 1). Our primary aim was to describe the development and psychometric properties of a novel measure on suicidal communications functions. The initial pool of items included in Study 1 (i.e., exploratory factor analysis) reflected each of the potential functions described above including help-seeking, plans and preparations, interpersonal emotion regulation, and ambivalence resolution. Amazon Mechanical Turk workers with a past-year history of suicide ideation completed a survey, and these data were analyzed using factor analysis and item response theory. Finally, we examined the internal consistency reliability and validity of the measure. The purpose of this study is to clarify the functions of suicidal communications to inform prevention and intervention efforts. 

## 3. Method

### 3.1. Participants

The current study examines the latent structure, internal consistency reliability, and validity of the communicant version of the Functions of Suicidal Communications Scale (FSC-C). A sample size of 500 or more is recommended for factor analysis [32,33,34]. Given the 35-item length of the original pool of items for the FSC-C, the recruitment goal for this study was a minimum of 700 people with past-year suicide ideation according to the SBQ-R, a minimum of 700 people with past-year exposure to suicide ideation. Participants were recruited via an Amazon Mechanical Turk (MTurk) Human Intelligence Task (HIT). Inclusion criteria were as follows: 18 years of age, 95% MTurk worker approval rating, English speaking and reading, and located in the United States. Workers meeting these criteria were eligible to review a consent form and indicate their consent to participate (yes/no: “I am 18 years of age or older and am willing to participate”). If eligible workers endorsed either past-year suicide ideation or exposure to suicide ideation or behavior (or both), completed the 30–45 min survey, and passed a selection of 5 attention checks, they received USD 1.00 for their time. Not all questions were required depending on endorsement of previous items. Qualtrics logic guided participants through the questions depending on their responses. For example, if a participant did not endorse past-year suicide ideation, they did not complete the remainder of the communicant form. A completion code system was used to ensure that each participant filled out the Qualtrics survey prior to dispersing payment [35,36]. This study was Institutional Review Board (IRB) approved by the University prior to beginning data collection. 

In total, 1338 MTurk workers successfully completed the 30–45 min anonymous survey, and of the final sample, 896 endorsed some form of suicide communication within the past year and completed the FSC-C. See Figure 2 for a survey administration flowchart. Participants identified as female (50.2%), male (48.3%), transgender (0.6%), gender non-conforming (0.3%) and other (0.5%). The majority identified as white (73.4%), followed by Black (11.3%), Hispanic or Latino (5.3%), Asian (5.3%), and other (4.7%). The majority also identified as straight (79.2%), followed by bisexual (14.9%), gay (3.5%), and other (0.14%, e.g., asexual and pansexual). Regarding relationship status, participants were married (54.2%), in a committed relationship (18.0%), single (22.3%), divorced (4.6%), and widowed (0.8%). Over half (55.8%) reported having children. Nearly half reported a psychiatric or mental health condition (44.5%) and attended psychological counseling one or more times within the past six months (45.8%). Regarding hospitalizations for suicide attempts, 26.4% reported one or more hospitalizations. The majority identified as employed and working full time (72.7%) and reported a household income of USD 50,000 or more (54.1%). Finally, the majority also identified as Christian or Catholic (63.8%), though other religious identities (e.g., Jewish; 2%; Muslim; 1.2%) as well as atheism and agnosticism (22.0%) were also represented. See Table 1 for a summary of sample demographics. 

### 3.2. Item Generation and Scale Development 

The original FSC-C item pool included 35 items developed to fit within a theoretical model of suicidal communications. There are two versions of the scale: one for recipients and the other for communicants. The items are identical in each of the versions, but worded differently for recipients and communicants (e.g., “I wanted to say goodbye to that person” and “They wanted to say goodbye”). The current study focuses on the structure of the communicant version (FSC-C). At the time these items were generated, there was no prior empirical research on the functions of suicidal communications and to date, there is no published validated measure on the topic. As a result, example items were generated based on related research (see the introduction, for a review) and subject matter expert review. Specifically, subject matter experts in measure development and suicide supervised the generation of items and construction of the battery. This measure was intended to be multidimensional with four hypothesized factors: help-seeking, interpersonal emotion regulation, preparations for suicide, and ambivalence resolution. Example items include “ I wanted to say goodbye”, and “I thought talking about how I was feeling would help me feel less overwhelmed”. The scale was preceded by the following prompt: “In the past year when you were thinking about suicide, why did you choose to communicate these thoughts? Indicate how well the responses below capture the reason you chose to communicate on a scale of 0 (Not at all) and 5 (Completely True). Base your responses on what you thought at the time even though you may think differently now.” Item ratings are on a 5-point Likert scale, with higher ratings indicating greater endorsement of the item. 

### 3.3. Screener Measures 

**Suicide ideation and behavior.** Participants’ past-year suicide ideation and behavior history was assessed using the Suicide Behavior Questionnaire-Revised (SBQ-R; Osman et al., 2001). This is a brief (5-item) measure that assesses history of suicide ideation, intent, and attempts, which served the primary purpose of a screener to exclude people without a past-year history of suicide ideation. However, the first two items of the scale were used to examine construct validity. Internal consistency was acceptable to good in prior samples, ranging from 0.76 to 0.87 [37]. The last question of this measure assesses the likelihood that the participant will attempt suicide in the future was removed for the current study due to the anonymous nature of the survey. The internal consistency reliability of the scale in the current sample is 0.805, indicating good reliability. Participants who endorsed any level of past-year suicide ideation were eligible to complete the communicant version of the battery. 

### 3.4. Validity 

**Motivations for Suicide Attempts.** Given the lack of prior measures on the functions or motivations of suicidal communications, motivations for suicide attempts were assessed. Participants’ motivations for suicide attempts were assessed using the Inventory for Motivations of Suicide Attempts (IMSA) [38]. This is a 54-item measure assessing interpersonal (e.g., “wanted to make people sorry for the way they treated me”) and intrapersonal (“my emotions were too overwhelming to handle”) motivations for suicide attempts. Internal consistency was acceptable in prior samples at 0.66 or higher [38]. In the current sample, the internal consistency reliability of the intrapersonal and interpersonal motivations scales was excellent at 0.935 and 0.952, respectively. 

### 3.5. Data Analysis 

**Overview.** Prior to conducting these analyses, the sample was randomly split in half for the EFA and CFA, respectively. An exploratory factor analysis (EFA) and parallel analysis were conducted on the original 35-item FSC-C in SPSS Version 26 to examine the latent structure of the measure on one half the sample. Using maximum likelihood estimation and Promax rotation, items with factor loadings above 0.50 and cross-loadings below 0.35 were retained. In addition to examining the latent structure of the scale and reducing the number of items, we examined internal consistency reliability and criterion validity. Using M*Plus* v.8 created by Muthen and Muthen [39], confirmatory factor analysis (CFA) was used on the second half of the sample to compare model fit indices between several alternative models: a correlated two-factor solution, unidimensional model, bifactor model, 7-item model prescribed by the bifactor model calculator, and a short 4-item version assembled using bifactor modeling and item response theory. Finally, an item response theory (IRT) graded response model (GRM) was conducted on the second half (CFA) sample using IRTPRO [40].

**Model fit.** To compare measurement model fit, v.8 was used to examine the following fit indices and recommended cut-offs (Kline, 2006): the comparative fit index (CFI ≥ 0.90), the Tucker Lewis Index (TLI ≥ 0.95), the root mean square error of approximation (RMSEA ≥ 0.06), the standardized root mean square residual (SRMS ≥ 0.08). The chi-square test statistic was also examined with non-significant values indicating good fit. However, given the large sample size, this test statistic is likely inaccurate [34]. Additional bifactor indices of model reliability and dimensionality were calculated using Dueber’s [41] excel-based workbook and interpreted using guidelines from Rodriguez, Reise, and Haviland [42].

## 4. Results 

### 4.1. Data Cleaning 

Before conducting our primary analyses, we examined data for missing values, univariate and multivariate outliers. There were no missing data as all questions were required to continue with the survey. The dataset was randomly split to conduct an EFA (*n* = 415) and a CFA(*n* = 433). 

In the total FSC-C sample, there were no univariate outliers, defined as observations having z-scores > 3. To identify multivariate outliers, the Mahalanobis distance was assessed for each participant utilizing the chi-square critical value of 65.25. There were 72 multivariate outliers (8.08% of the total sample). Following the recommendation of Meyers, Gamst, and Guarino [43] to delete outliers exceeding 3% of the data, these outlier cases were deleted. Our data cleaning yielded a final sample of 819 participants who completed the FSC-C measure. With regard to skew and kurtosis, none of the values exceeded +/− 2 or +/− 9, respectively, indicating that transformation of the data was not needed [44]. 

### 4.2. Parallel Analysis

A parallel analysis was conducted on the original set of 35 items. Parallel analysis adjusts for the effect of sampling error inherent in the K1 method and therefore provides a sample-based alternative to the population-based K1 criterion [45,46]. According to Hayton and colleagues [47] factors corresponding to actual eigenvalues that are greater than the parallel average random eigenvalues should be retained. The parallel analysis revealed that the eigenvalues for the sample correlation matrix exceeded the average eigenvalues for two factors, explaining 71.40% of the variance (i.e., 6.54 and 1.35, respectively). Therefore, two factors were retained.

### 4.3. Exploratory Factor Analysis 

An exploratory factor analysis (EFA) of the 35-item Functions of Suicidal Communications-Communicant Version (FSC-C) scale was performed restricted to two factors on data from 415 MTurk workers utilizing IBM SPSS Version 26.0 [48]. The Kaiser-Meyer-Olkin measure of sampling adequacy was 0.948, indicating that the present data were suitable for exploratory factor analysis. Similarly, Bartlett’s test of sphericity was significant (*p* < 0.001), indicating sufficient correlation between the variables to proceed with the analysis. As recommended by Tabachnik & Fiddell [49], we assumed that the factors were correlated and therefore extracted two factors with a Promax rotation and maximum likelihood estimation. In the original pool of items, the two factors were moderately correlated (*r* = 0.532). Examination of the scree plot showed the last visible drop occurring between the second and third factors, thus further supporting the results of the previous parallel analysis.

After removing items with loadings below 0.5 and cross-loadings below 0.35, the resulting two factors accounted for 64.4% of the total variance. Of the original 35 items, 15 were retained. The structure coefficients from the Promax rotation are presented in Table 2. The magnitude of the structure coefficients was moderate to large for all the items. Subscales were constructed as follows: Factor 1 (items: 11, 18, 19, 26, 28, 30, 32, 33, and 34) and Factor 2 (9, 12, 13, 15, 16, 17, and 27). The internal consistency reliability of each subscale as assessed by Cronbach’s coefficient alpha is reported in Table 3. Both subscales exhibited good to excellent internal consistency reliability. Factor 1 consisted of items related to planning and preparatory behavior as well as interpersonal conflict and Factor 2 consisted of items related to coping and emotional relief *(r* = 0.293). 

### 4.4. Confirmatory Factor Analysis (Correlated Factors, Unidimensional, and Bifactor Model) 

Using maximum likelihood estimation, the correlated two-factor model was specified as follows: Factor 1 (items: 11, 18, 19, 26, 28, 30, 32, 33, and 34) and Factor 2 (9, 12, 13, 15, 16, 17, and 27). The chi-square test was significant, indicating that the model was not a perfect fit for the data (𝝌^2^(103) = 746.30, *p* < 0.001). However, the chi-square statistic is highly sensitive to sample size (*n* = 433) [50]; and may not be an accurate representation of model fit. This model otherwise yielded poor fit indices: CFI (0.73), TLI (0.69), RMSEA (0.12), and SRMR (0.12). Further, Factor 1 explained most of the variance (83.4%), and the two factors were highly correlated (*r* = 0.99) indicating a need to compare a unidimensional with a bifactor model solution [51]. Table 2 presents the standardized factor loadings from the CFA, which were low to moderate in magnitude. Given the two-factor solution suggested by the EFA and parallel analysis and these poor fit indices, we ran a unidimensional and bifactor model to compare the fit indices.

Similar to the two-factor model, the chi-square test was significant for the bifactor model (𝝌^2^(88) = 322.68, *p* < 0.001). The model otherwise yielded borderline to acceptable indices of fit: CFI (0.90), TLI (0.87), RMSEA (0.08), and SRMR (0.08). Regarding the unidimensional model, the chi-square test was also significant (𝝌^2^(88) = 747.42, *p* < 0.001). Moreover, the model yielded poor fit indices: CFI (0.73), TLI (0.70), RMSEA (0.12), and SRMR (0.13). Therefore, when comparing the three models, the bifactor model evidenced the best fit and ancillary bifactor indices of model reliability and dimensionality were calculated [42]. These indices were calculated using an excel-based workbook created by Dueber [41]. 

These ancillary indices suggested that the general factor was strong, explaining 72.3% of the common variance and between 35.7% and 96.2% of the total variation in item responses (see Table 4 for individual ECV values). Items 11, 28, 30, 26, 13, 15, and 9 were most representative of the general dimension, and were selected for inclusion in the final measurement model. The model-based omega reliability was excellent for the general factor (ω = 0.918) as well as Factor 1 (ω = 0.87) and Factor 2 (ω = 0.83). Additional evidence for the relative strength of the general factor was apparent in its percent of reliable total score variance (ω*H*/ω = 0.87), suggesting that 87.0% of reliable variance in the multidimensional FSC-C model was explained by this general factor. The percent of reliable variance for Factors 1 and 2 was low at 13.1% and 0.40%, respectively, after removing variation explained by the general factor. 

The 7-item model prescribed by the bifactor model calculator yielded a superior fit relative to prior models. These items were selected due to having an individual explained common variance (IECV) value of 0.80 or higher (See Table 4), suggesting the general factor explained 80% or more of the variance in each item. Such items are most representative of the content of the general dimension and should be selected for inclusion in a unidimensional model [52]. As with the other models, the chi-square test was significant, indicating that the model was not a perfect fit for the data (𝝌^2^(14) = 52.64, *p* < 0.001). However, as previously stated, chi-square statistic is highly sensitive to sample size (*n* = 448) and may not be an accurate representation of model fit. This model otherwise yielded acceptable fit indices: CFI (0.94), TLI (0.90), RMSEA (0.08), and SRMR (0.04). However, this model explained only 51.8% of the variance. The standardized factor loadings were moderate to high in magnitude. Given the essential unidimensional structure, item response theory was used to examine the properties of individual items, and to finalize the scale by choosing those items that best represented the latent construct.

### 4.5. Item Response Theory 

Using a graded response model (GRM), the information and discrimination levels were examined for the 7 items of the FSC-C recommended by the bifactor model (see Table 5). The cut-off for information level was 2.00. Of the original 7-item pool, 3 items were selected for removal due to not providing sufficient information about the latent construct: 9, 13, and 15. The remaining items (11, 26, 28, and 30) were examined for their “difficulty” level with values less than zero indicating greater than 50% probability that a response option will be selected. Examining all items’ difficulty levels revealed that low to moderate levels of the latent trait were more likely to be endorsed across items. Figure 3 displays the information curve for each item. 

### 4.6. Confirmatory Factor Analysis (4-Item Short Form) 

A final CFA was run using the IRT and Bifactor model prescribed items. That is, there were four items recommended by the bifactor model calculator (11, 26, 28, and 30), and IRT GRM model. The final model was chosen based on items that were both recommended by the bifactor model calculator and provided sufficient information for the IRT GRM model. For this 4-item measure, the chi-square test was not significant, indicating that the model a good fit for the data (𝝌^2^(2) = 1.81, *p* = 0.40). This model yielded perfect fit indices: CFI (1.00), TLI (1.00), RMSEA (0.00), and SRMR (0.01). Table 6 compares fit indices across models and demonstrates the superior fit of the final item selection using IRT and the bifactor model calculator (See Table 6). Figure 4 displays the final measurement model. Table 7 displays a comparison of model fit. 

### 4.7. Criterion Validity 

Given the lack of another measure on the functions of suicidal communications, there was no available measure with which to directly compare the FSC-C. As a result, these analyses are limited to divergent validity to demonstrate that the functions of suicidal communications are a distinct construct from suicide ideation and behavior, a single item on suicidal communications (item 2 of the SBQ-R), and interpersonal and intrapersonal motivations for suicide attempts. 

The first item of the SBQ-R (i.e., “Have you ever thought about or attempted to kill yourself?) was significantly and positively correlated with the FSC-C (*r* = 0.09, *p* = 0.006). This significant but low correlation suggests that endorsement of ambivalence motivations is associated with having some history of suicide ideation and/or behavior, but it is not a strong indicator of the severity of the history. To that point, the correlation between the ambivalence resolution function and the second item of the SBQ-R (i.e., “How often have you thought about killing yourself in the past year?”) was negative and not significant (*r* = −0.05, *p* = 0.872). This result suggests that there is no significant association between ambivalence motivations and past-year suicide ideation frequency/severity, but that overall being motivated by ambivalence resolution may indicate less severe/frequent ideation. The third item from the SBQ-R on suicidal communications was also significantly and positively correlated with the FSC-C (*r* = 0.20, *p* < 0.001). The fourth item was omitted due to the anonymous nature of the survey. 

The intrapersonal motivations for suicide scale was significantly and positively associated with the FSC-C (*r* = 0.45, *p* < 0.001). The interpersonal motivations for suicide scale was also significantly and positively associated with the FSC-C (*r* = 0.72, *p* < 0.001). These results indicate that ambivalence resolution motivations for suicidal communications are highly associated with interpersonal motivations for suicide. Given that the interpersonal motivations scale of the IMSA encapsulates help-seeking and interpersonal influence motivations, the motivations/functions described in both scales are highly similar but still distinct constructs. 

## 5. Discussion 

The current study was a psychometric pilot of a novel measure written to assess four potential functions of suicidal communications. Developed to fit within a theory [53] that synthesizes existing empirical work on suicidal communication, the original set of items examined the potential for multiple functions of suicidal communications. Specifically, we posited four motivations: help-seeking, preparatory behavior, interpersonal emotion regulation, and ambivalence resolution. Results did not confirm the multiple functions of suicidal communications hypothesis. They revealed a unidimensional structure, capturing ambivalence resolution motivations. As the first measure to assess this construct, the current study represents a first step towards a comprehensive and theory-based measurement of suicidal communications. Results illuminate the next steps required for rigorous assessment of suicidal communication functions. 

Given limited theoretical and empirical work [6,7], the current study relied on quantitative indices over theory-based measure development strategies to test the multiple functions of suicidal communications hypothesis. Item response theory methods were used following exploratory and confirmatory factor analysis to assess the measure’s structure. Initial signs (e.g., parallel analysis and EFA) pointed to a two-factor solution describing preparatory behavior and interpersonal conflict-related motivations, and IER motivations. Due to high correlations between both factors and the small percent of variance explained by the second factor, bifactor modeling was an appropriate next step in the analytic plan. Although the model converged, ancillary indices of fit and reliability indicated that the measure was primarily unidimensional. 

The resulting scale comprised items related to final attempts to get help, indirect or avoidant communication, seeking connection, and suicide decision making. Given that people may communicate about suicide to strengthen relationships, get help or support [12], or say goodbye [23], these items primarily reflect an ambivalence resolution function that may encapsulate the undecided nature of most suicide decision making. While considering unsolidified suicide plans and preparations, the same individual may also be seeking help and relationship improvement. Given that people at risk of suicide experience low connectedness [54], seeking conflict resolution or relationship repair may be a form of ambivalence resolution and reflect a critical part of the suicide decision-making process [30]. 

Results suggest that there may not be categorically different functions of suicidal communications. Instead, ambivalence may underscore these seemingly disparate functions (help-seeking, preparation, interpersonal emotion regulation), characterized by conflicting desires to seek help or increase one’s connection to the world, while at the same time, considering preparation for death. Given the evidence that ambivalence about living and dying (as opposed to certainty about a desire to die) characterizes the suicidal state [55,56], communication during this period is likely to reflect this internal struggle. Further, since ambivalence about suicide comprises the simultaneous wish to live and the wish to die [57], we might expect measure items such as those retained for the current study to reflect both dimensions. Finally, the role of ambivalence in suicidal communication is consistent with the non-linear change processes of suicide risk and desire (in addition to stable constructs) proposed by the fluid vulnerability theory of suicide [58]. As suicidal desire and risk dynamically fluctuate in response to external and internal processes throughout periods of acute and baseline risk, communication about suicide may express this fluctuation by simultaneously serving life-oriented and death-oriented purposes. 

If future research supports the ambivalence resolution function of suicidal communications, then communication may serve a key role in suicide decision making. They may bridge the “mulling over” and preparations steps on the path to suicide attempts [30]. People may communicate thoughts of suicide to prepare for death and determine if they want to go through with an attempt. The outcome of suicidal communications may provide information relevant to a person’s decision-making process (e.g., improving a relationship and obtaining needed help after several attempts). Contrary to communication motivations for suicide attempts [59], ambivalence resolution motivations for suicidal communications may, therefore, be associated with greater risk. Lengthier decision-making periods are associated with increased risk [31,60], and suicidal communications occurring during these periods may indicate a potentially critical point of transition from ideation to action. 

Notably, the function of ambivalence resolution, as measured by the current scale, was positively and significantly correlated with history of suicide ideation and behavior, though the correlation was low. There was alternatively no significant association between past-year suicide ideation severity/frequency and ambivalence functions. Prior research indicates an association between reduced or non-communication (versus communication) and elevated suicide risk [4,61,62]. In some respects, this non-significant correlation confirms prior work by finding a non-significant negative correlation with past-year ideation severity. However, the positive and significant correlation with lifetime suicide ideation and behavior suggests that people endorsing ambivalence motivations may have a history of heightened risk. These findings do not necessary suggest that those who communicate are at more or less risk of suicide, but only that people who endorse ambivalence resolution motivations are more likely to have experienced more of the suicide ideation and behavior spectrum to warrant a need for ambivalence resolution. Endorsement of ambivalence resolution motivations, therefore, suggests at least having deeply considered suicide as an option. Finally, as expected, the ambivalence resolution function was highly correlated with interpersonal motivations for suicide attempts in comparison with intrapersonal motivations. However, this correlation was not high enough to suggest that the two scales measured identical constructs. People engaging in suicidal behavior who report help-seeking and interpersonal motivations may also be more likely to communicate their intentions to resolve ambivalence. 

## 6. Limitations 

The present study is a first attempt at operationalizing the functions of suicidal communications. Given that the FSC-C is the first measure on the construct, the current study is also the first to test the multiple functions hypothesis. However, several limitations must be considered when interpreting our results. First, participants were recruited from Amazon Mechanical Turk (MTurk), eliciting potential data quality concerns. We took several steps to address this risk, such as the 95% approval rating eligibility criterion [63], removal of duplicate respondents, free response and Likert scale attention checks, and a captcha to enter the survey. Further, we adhered to published pay recommendations available at the time the data were collected [35]. However, these measures only reduced rather than eliminated the risk of collecting inattentive or fake respondents.

In total, over 50% of the data were removed due to low quality based on a series of attention checks and not utilized in the current study. However, recent research suggests that MTurk data quality may have declined in recent years [64] and researcher perception of fair compensation may not match participant expectations [65]. MTurk best practices research must stay abreast of the changing pay expectations and data quality assurance practices. As a result, low-quality responses may have eluded our careful review though all best practices available by the data collection start date were implemented.

Additionally, MTurk participants may not have represented the spectrum of suicidality as well as a clinical sample. Despite this limitation, several studies indicate that MTurk workers may yield a sample with less bias and more diversity than laboratory samples [36,66]. Further, MTurk workers report higher depression rates than other types of convenience samples, such as undergraduate college students [67]. Compared with other convenience samples, MTurk may have advantages for recruiting participants with suicide ideation and behavior but may not yield the advantages of a clinical sample. 

Most importantly, although the original item pool was developed based on prior empirical and theoretical work, existing research in this area is limited. The current study is a preliminary step towards formally operationalizing the functions of suicidal communications. Therefore, the resulting unidimensional measure may indicate the limited prior research on which to base item generation than the latent structure of suicidal communications. Relatedly, since there was no measure on the topic (i.e., the functions of suicidal communications) with which to compare psychometric properties, validity indices were limited to suicide ideation and behavior, and motivations for suicide attempts. Prior scales on suicidal communications focus on the helpfulness of family reactions and the amount of detail included in disclosures, neither of which assess functions of communications. While these scales are focused on suicidal communications, they are entirely different constructs than the one we aimed to capture. The scale focused on depth of disclosure [10], which is the only validated measure on suicidal communications, was, unfortunately, not included in the survey due to system error. 

## 7. Conclusions

There may be one function of suicidal communications. Disclosure may elicit connection and reasons for living that serve as barriers to suicide and resolve ambivalence. Future research should focus on engaging those with lived experience of suicide through interviews, focus groups, and participatory action methods. Such methods will aid in generating new items, revising current items, and obtaining feedback on theoretical models. Future studies should pilot a revised measure in clinical and community samples following the generation of additional items. The current measure serves as a starting point for the comprehensive measurement of suicidal communications and suggests that ambivalence resolution may be the sole function of suicidal communications. However, it may also merely suggest that the authors oversampled this latent construct. Engaging research participants and partners with lived experience may suggest functions not included in the current study or in prior research.’

## Figures and Tables

**Figure 1 ijerph-19-10081-f001:**
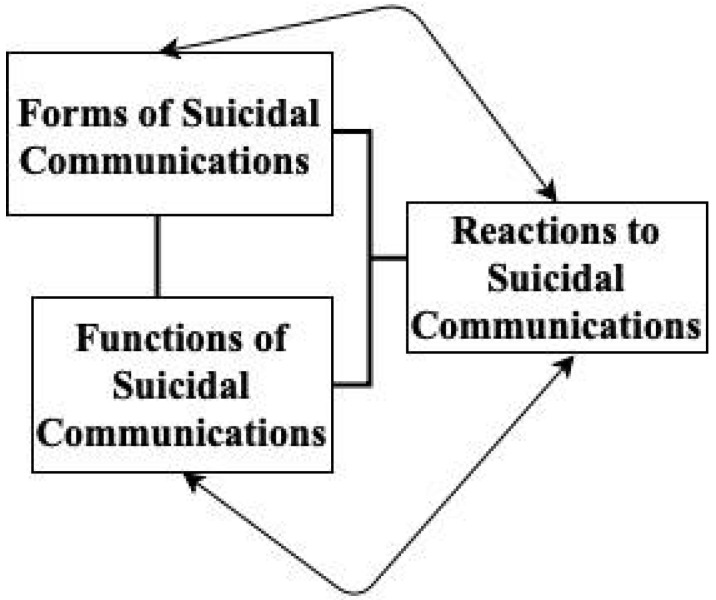
Preliminary empirical framework for suicidal communications.

**Figure 2 ijerph-19-10081-f002:**
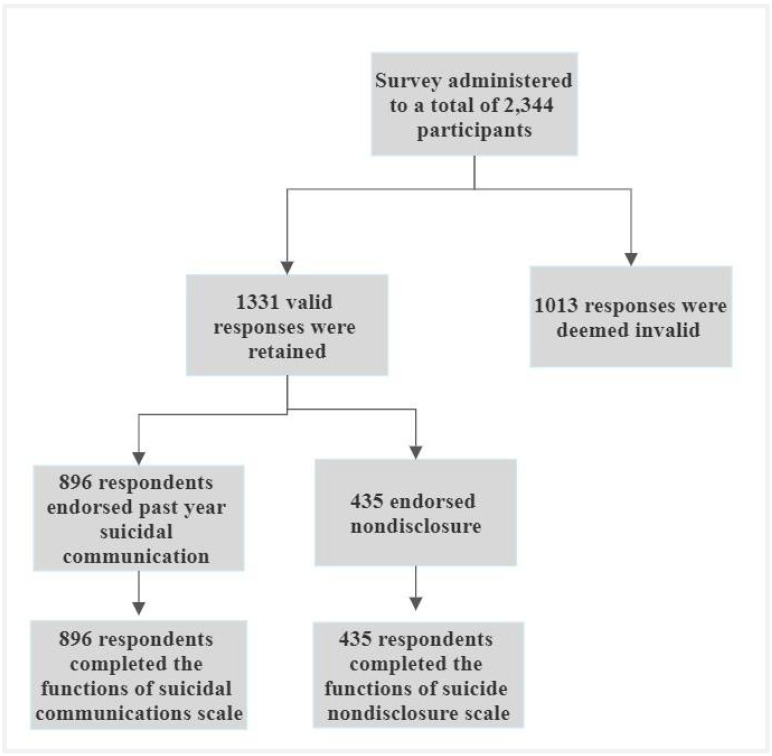
Survey administration flowchart.

**Figure 3 ijerph-19-10081-f003:**
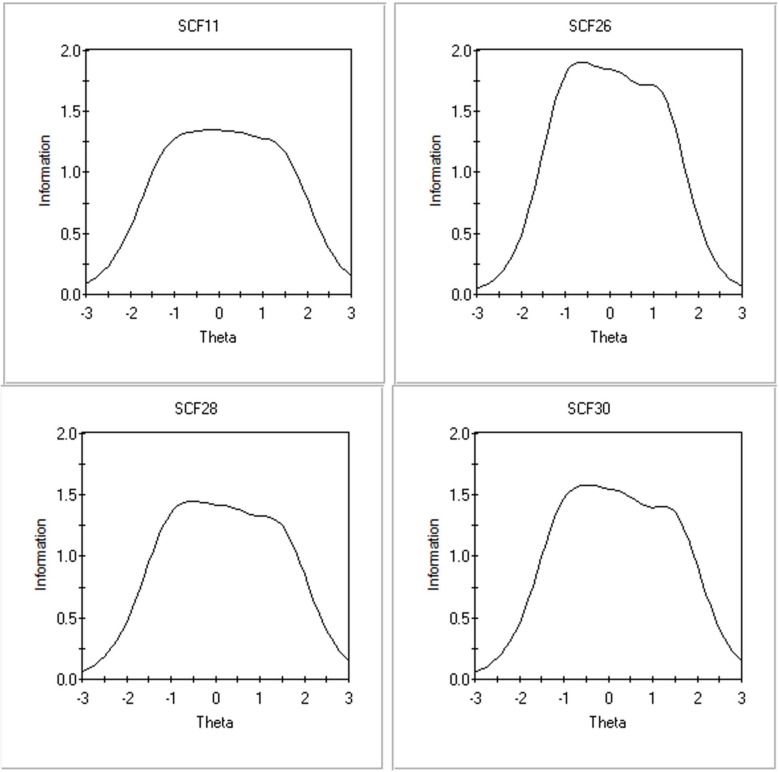
Item information curves for items 11, 26, 28, and 30.

**Figure 4 ijerph-19-10081-f004:**
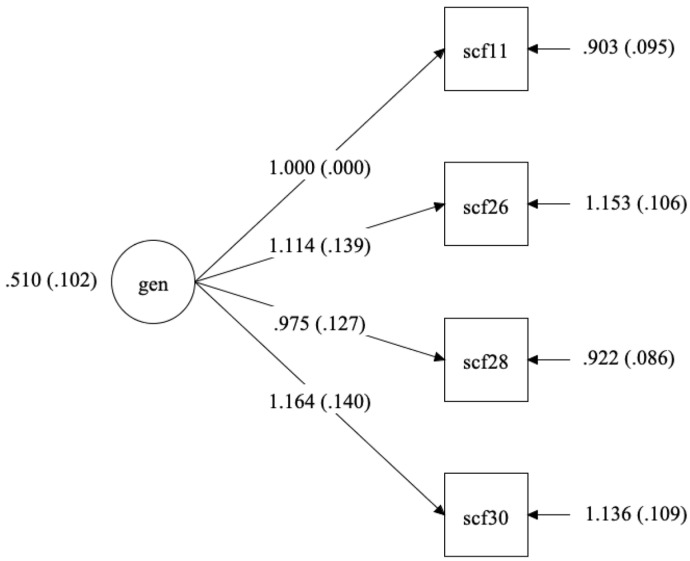
Functions of Suicidal Communications Scale—Communicant Version, 4-item measurement model assessing the ambivalence resolution function.

**Table 1 ijerph-19-10081-t001:** Sample demographics (*N* = 895).

Variable	*%*
**Gender Identity**	
Male	50.2
Female	48.3
Transgender	0.6
Gender Non-Conforming	0.5
**Race/Ethnicity**	
White/Caucasian	73.4
Black/African American	11.3
Hispanic or Latino	5.3
Asian	5.3
Other	4.7
**Sexual Identity**	
Straight/Heterosexual	79.2
Bisexual	14.9
Gay/Homosexual	3.5
Other	0.14
**Relationship Status**	
Married	54.2
Committed Relationship	18.0
Single	22.3
Divorced	4.6
Widowed	0.8
**Children**	55.8
**Mental Health Condition**	44.5
**Counseling (Past 6 Mo)**	45.8
**Hospitalization(s)/SA**	26.4
**Employed/Working Full Time**	72.7
**Household Income USD 50K^+^**	54.1
**Religion**	
Christian/Catholic	63.8
Atheism/Agnosticism	22.0
Jewish	2.0
Muslim	1.2

**Table 2 ijerph-19-10081-t002:** Structure coefficients for retained FSC-C items from the EFA and standardized loadings from the two-factor CFA.

	EFA		CFA Correlated Factors	
Item Number and Wording	1	2	1	2
11. It was my final attempt to get help	0.749	0.284	0.525	-
18. I needed someone to help me get my affairs in order	0.746	0.240	0.415	-
19. I wanted to say goodbye	0.789	0.222	0.670	-
26. I believed they would help me make my own decision and support me either way	0.703	0.302	0.712	-
28. I wanted to avoid an immediate response or have a way out of the conversation	0.767	0.212	0.500	-
30. I wanted to get closer to the person I told	0.734	0.306	0.582	-
32. I wanted the person I told to realize how much they hurt me	0.760	0.233	0.662	-
33. I wanted to cause the person I told pain	0.838	0.150	0.653	-
34. I was in a conflict with someone and thought it would solve the problem	0.814	0.112	0.663	-
9. I wanted someone to simply listen	0.048	0.625	-	0.619
12. I thought talking about how I was feeling would help me feel less overwhelmed	0.279	0.720	-	0.677
13. I was trying to cope with how I was feeling	196	0.775	-	0.469
15. I wanted emotional relief	0.183	0.767	-	0.633
16. I wanted to get it off my chest	0.259	0.721	-	0.420
17. I wanted someone to be aware of how I was feeling	0.248	0.726	-	0.483
27. I wanted the person I told to be aware of what I was going through	0.304	0.763	-	0.655

Note: exploratory factor analysis (EFA), Functions of Suicidal Communications Scale—Communicant Version (FSC-C), and confirmatory factor analysis (CFA).

**Table 3 ijerph-19-10081-t003:** Internal Consistency of Correlated 2 Factor Items.

	Cronbach Alpha
Factor 1	0.928
Factor 2	0.870

**Table 4 ijerph-19-10081-t004:** FSC-C standardized loadings and individual explained common variance in the bifactor model.

Item Number and Wording	Gen.	F1	F2	IECV
**11. It was my final attempt to get help**	**0.596**	**−0.147**	-	**0.943**
18. I needed someone to help me get my affairs in order	0.546	0.280	-	0.792
19. I wanted to say goodbye	0.584	0.307	-	0.783
**26. I believed they would help me make my own decision and support me either way**	**0.662**	**0.201**	-	**0.916**
**28. I wanted to avoid an immediate response or have a way out of the conversation**	**0.620**	**0.212**	-	**0.884**
**30. I wanted to get closer to the person I told**	**0.597**	**0.058**	-	**0.991**
32. I wanted the person I told to realize how much they hurt me	0.573	0.414	-	0.657
33. I wanted to cause the person I told pain	0.597	0.610	-	0.425
34. I was in a conflict with someone and thought it would solve the problem	0.519	0.697	-	0.357
**9. I wanted someone to simply listen**	**0.619**	-	**−0.122**	**0.962**
12. I thought talking about how I was feeling would help me feel less overwhelmed	0.626	-	−0.368	0.743
**13. I was trying to cope with how I was feeling**	**0.567**	-	**0.278**	**0.806**
**15. I wanted emotional relief**	**0.608**	-	**−0.123**	**0.961**
16. I wanted to get it off my chest	0.554	-	0.474	0.577
17. I wanted someone to be aware of how I was feeling	0.604	-	0.479	0.614
27. I wanted the person I told to be aware of what I was going through	0.608	-	−0.336	0.766

Note: confirmatory factor analysis (CFA); Functions of Suicidal Communications Scale—Communicant Version (FSC-C). Bolded items indicate those with standardized loadings and IECV values above the cut-off for inclusion in the final model.

**Table 5 ijerph-19-10081-t005:** IRT parameters for the 7 FSC-C items from the bifactor model.

Item	Mean	*SD*	*a*	*b1*	*b2*	*b3*	*b4*
9	3.05	0.972	0.16	−19.56	−12.95	−5.25	5.24
**11**	**1.96**	**1.30**	**2.37**	**−0.96**	**−0.23**	**0.47**	**1.22**
13	2.92	1.04	0.54	−5.19	−3.77	−1.08	2.33
15	2.86	1.01	0.55	−5.49	−3.55	−1.31	1.93
**26**	**2.13**	**1.34**	**2.35**	**−0.94**	**−0.26**	**0.39**	**1.53**
**28**	**1.97**	**1.31**	**2.60**	**−0.93**	**−0.26**	**0.50**	**1.48**
**30**	**1.98**	**1.30**	**2.58**	**−0.88**	**−0.27**	**0.41**	**1.35**

Note: item response theory (IRT), Functions of Suicidal Communications Scale—Communicant Version (FSC-C), and exploratory factor analysis (EFA). Bolded items indicate those with information levels above the cut-off for inclusion in the final model.

**Table 6 ijerph-19-10081-t006:** FSC-C standardized loadings, final 4-item unidimensional model.

Item Number and Wording	Gen.
11. It was my final attempt to get help	0.596
26. I believed they would help me make my own decision and support me either way	0.662
28. I wanted to avoid an immediate response or have a way out of the conversation	0.620
30. I wanted to get closer to the person I told	0.597

Note: confirmatory factor analysis (CFA); Functions of Suicidal Communications Scale—Communicant Version (FSC-C).

**Table 7 ijerph-19-10081-t007:** Comparative confirmatory factor analysis model fit statistics.

FSC-C Model	𝝌^2^	*df*	CFI	TLI	RMSEA	SRMR
Correlated factors	747.29	103	0.73	0.69	0.12	0.12
Bifactor	322.68	88	0.90	0.87	0.08	0.08
Unidimensional	747.42	88	0.73	0.69	0.12	0.12
7-item	52.64	14	0.94	0.90	0.08	0.04
4-item	1.81	2	1.0	1.0	0.00	0.01

Note: confirmatory factor analysis (CFA); Functions of Suicidal Communications Scale—Communicant Version (FSC-C).

## Data Availability

The data presented in this study are available on request from the corresponding author. The data are not publicly available to protect confidentiality and privacy.

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
