# Peer review of "A Psychometric Pilot Study Examining the Functions of Suicidal Communications Using IRT and Factor Analysis"

_ijerph, 2022, doi:10.3390/ijerph191610081_

Round 1

Reviewer 1 Report

The study has a relevant and current theme. It presents an approach that can contribute to prevention actions, by identifying individuals who are at risk. The study was well conducted, the argumentation is coherent, and the analyses were sufficiently described. I have only a few observations. The abstract does not accurately express the content of the article. The purpose of the study needs to be made more evident. The methods described do not include the data analysis technique. And the results are limited and relevant findings have been omitted. Also, acronyms were used that are not clear in the abstract. 

The methods should include a flowchart describing the sampling and application of the instrument. 

In the results include a table to present the characteristics of the population studied. 

The study has many results, in some moments the comprehension was not easy. However, the discussion was too brief. The limitations of the study are about the same size as the discussion. 

Author Response

The study has a relevant and current theme. It presents an approach that can contribute to prevention actions, by identifying individuals who are at risk. The study was well conducted, the argumentation is coherent, and the analyses were sufficiently described. I have only a few observations.

The abstract does not accurately express the content of the article. The purpose of the study needs to be made more evident. The methods described do not include the data analysis technique. And the results are limited and relevant findings have been omitted. Also, acronyms were used that are not clear in the abstract.

Thank you for your helpful recommendations for the abstract. The authors agree that the abstract was too brief and did not include the important information you listed. The abstract, on page 2 of the manuscript, is now updated, but also included directly below, italicized.

Background: Suicide prevention relies on suicidal communications to identify and intervene with those at risk. Scant research tests explicit theories of suicidal communication impeding applications to prevention science. The current study pilots a new measure assessing the functions of suicidal communications using factor analysis and item response theory factor analysis and item response theory. Methods: MTurk workers (n=898) completed an anonymous survey. The original scale included 35 items refined using exploratory and confirmatory factor analysis, bifactor modeling, and item response theory graded response models.  Results: The initial EFA identified a two correlated factor solution. The two correlated 2-factor and unidimensional models yielded poor fit. A bifactor model yielded borderline to acceptable fit. The final The final 4 items scale were identified using a bifactor model and item response theory graded response models and included was 4 items capturing ambivalence resolution defined as behaviors aimed to aid in suicide decision-making. The final model yielded excellent fit: ?2(2)=1.81, CFI (1.00), TLI (1.00), RMSEA (.00), and SRMR (.01). Conclusions: There may be one function of suicidal communications. Disclosure may elicit connection and reasons for living that serve as barriers to suicide and resolve ambivalence. Key limitations include convenience sampling and limited validity measures. Future research should partner with participants to improve scale and theory development efforts.

The methods should include a flowchart describing the sampling and application of the instrument.

Thank you for this excellent suggestion. A flowchart, now labeled Figure 1, is now included in the manuscript and below for ease of viewing. It includes all of the information we have available regarding survey administration.

In the results include a table to present the characteristics of the population studied.

Thank you for this important suggestion. Please see below for a study sample demographics table.

Table 1. Sample demographics (N=895)

Variable

                        %

Gender Identity

Male

50.2

Female

48.3

Transgender

.6

Gender Nonconforming

.5

Race/Ethnicity

White/Caucasian

73.4

Black/African American

11.3

Hispanic or Latino

5.3

Asian

5.3

Other

4.7

Sexual Identity

Straight/Heterosexual

79.2

Bisexual

14.9

Gay/Homosexual

3.5

Other

.14

Relationship Status

Married

54.2

Committed relationship

18.0

Single

22.3

Divorced

4.6

Widowed

.8

Children

55.8

Mental Health Condition

44.5

Counseling (Past 6 Mo)

45.8

Hospitalization(s)/SA

26.4

Employed/Working Full-Time

72.7

Household Income $50K+

54.1

Religion

Christian/Catholic

63.8

Atheism/Agnosticism

22.0

Jewish

2.0

Muslim

             1.2

The study has many results, in some moments the comprehension was not easy. However, the discussion was too brief. The limitations of the study are about the same size as the discussion.

Thank you for this important suggestion. Due to the already extensive length of the paper, (40 pages) we elected to keep the discussion as brief as possible and also wanted to ensure all of the limitations were covered. If the reviewer wishes to identify specific topics that the discussion neglected, we can ensure they are included.

Reviewer 2 Report

First of all, congratulations for the work carried out, it is a very interesting research and the methodology of the study is very well defined and developed. After reviewing the manuscript, it could be improved in the following points:

The summary, from line 8 to 11, should be modified, it does not contextualize suicide as a serious public health problem and the need for preventive interventions.

Similarly, the introduction does not justify the study well. The introduction, line 21 through 61, should contextualize suicidal behavior as a serious public health problem and the need for preventive interventions. This enhances the rationale and interest of the research.

Best regards.

Author Response

Reviewer 2.

First of all, congratulations for the work carried out, it is a very interesting research and the methodology of the study is very well defined and developed. After reviewing the manuscript, it could be improved in the following points:

Thank you for taking the time to review and improve our manuscript and for your kind comments on the strengths and weaknesses of the study.

The summary, from line 8 to 11, should be modified, it does not contextualize suicide as a serious public health problem and the need for preventive interventions.

Thank you for this important suggestion. The revised abstract is below.

Background: Suicide prevention, an important public health issue, relies on suicidal communications to identify and intervene with those at risk. Scant research tests explicit theories of suicidal communication impeding applications to prevention science. The current study pilots a new measure assessing the functions of suicidal communications using factor analysis and item response theory. Methods: MTurk workers (n=898) completed an anonymous survey. The original scale included 35 items refined using exploratory and confirmatory factor analysis, bifactor modeling, and item response theory. Results: The initial EFA identified a two correlated factor solution. The correlated 2-factor and unidimensional models yielded poor fit. A bifactor model yielded borderline to acceptable fit.  The final 4 items were identified using a bifactor model and item response theory graded response models capturing a single ambivalence resolution factor, which is defined as behaviors aimed to aid in suicide decision-making. The final model yielded excellent fit: ?2(2)=1.81, CFI (1.00), TLI (1.00), RMSEA (.00), and SRMR (.01). Conclusions: There may be one function of suicidal communications. Disclosure may elicit connection and reasons for living that serve as barriers to suicide and resolve ambivalence. Key limitations include convenience sampling and limited validity measures. Future research should partner with participants to improve scale and theory development efforts.

Similarly, the introduction does not justify the study well. The introduction, line 21 through 61, should contextualize suicidal behavior as a serious public health problem and the need for preventive interventions. This enhances the rationale and interest of the research.

Thank you for this excellent suggestion. We included several references in the abstract, introduction, and discussion to contextualize suicidal behavior as a serious public health problem and how the current study was intended to inform prevention and intervention research. Suicidal communication is the necessary precursor to intervention so improving our understanding of the functions, forms, and reactions to suicidal communications is key. Please see the revised manuscript for tracked changes additions.
